# DEEP ENCODER, SHALLOW DECODER: REEVALUATING NON-AUTOREGRESSIVE MACHINE TRANSLATION

**Jungo Kasai**[♡][*]  **Nikolaos Pappas**[♡]  **Hao Peng**[♡]  **James Cross**[♣]  **Noah A. Smith**[♡♦]
[♡]Paul G. Allen School of Computer Science & Engineering, University of Washington
[♣]Facebook AI  [♦]Allen Institute for AI
`{jkasai,npappas,hapeng,nasmith}@cs.washington.edu`
`jcross@fb.com`

## ABSTRACT

Much recent effort has been invested in *non-autoregressive* neural machine translation, which appears to be an efficient alternative to state-of-the-art *autoregressive* machine translation on modern GPUs. In contrast to the latter, where generation is sequential, the former allows generation to be parallelized across target token positions. Some of the latest non-autoregressive models have achieved impressive translation quality-speed tradeoffs compared to autoregressive baselines. In this work, we reexamine this tradeoff and argue that autoregressive baselines can be substantially sped up without loss in accuracy. Specifically, we study autoregressive models with encoders and decoders of varied depths. Our extensive experiments show that given a sufficiently deep encoder, a *single-layer* autoregressive decoder can substantially outperform strong non-autoregressive models with comparable inference speed. We show that the speed disadvantage for autoregressive baselines compared to non-autoregressive methods has been overestimated in three aspects: suboptimal layer allocation, insufficient speed measurement, and lack of knowledge distillation. Our results establish a new protocol for future research toward fast, accurate machine translation. Our code is available at `https://github.com/jungokasai/deep-shallow`.

## 1 INTRODUCTION

Fast, accurate machine translation is a fundamental goal with a wide range of applications both in research and production. State-of-the-art neural machine translation systems generate translations *autoregressively* where words are predicted one-by-one conditioned on all previous words (Kalchbrenner & Blunsom, 2013; Sutskever et al., 2014; Bahdanau et al., 2015; Wu et al., 2016; Vaswani et al., 2017). This sequential property limits parallelization, since multiple tokens in each sentence cannot be generated in parallel. A flurry of recent work developed ways to (partially) parallelize the decoder with *non-autoregressive* machine translation (NAR; Gu et al., 2018), thereby speeding up decoding during inference. NAR tends to suffer in translation quality because parallel decoding assumes conditional independence between the output tokens and prevents the model from properly capturing the highly multimodal distribution of target translations (Gu et al., 2018).

Recent work proposed methods to mitigate this multimodality issue, including iterative refinement (e.g., Lee et al., 2018; Ghazvininejad et al., 2019), and modeling with latent variables (e.g., Ma et al., 2019; Shu et al., 2020). These approaches modify the decoder transformer to find a balance between decoding parallelism and translation quality. In this work, however, we adopt a different speed-quality tradeoff. Recent work by Kim et al. (2019) in autoregressive machine translation (AR) suggests that better speed-quality tradeoffs can be achieved by having different depths in the encoder and the decoder. Here, we make a formal argument in favor of *deep encoder, shallow decoder* configurations and empirically demonstrate better speed-quality tradeoffs for the AR baselines.

---

[*] Work partially done at Facebook AI.

We provide extensive speed-quality comparisons between iterative NAR models and AR models with varying numbers of encoder and decoder layers. In particular, we use two types of speed measures for translation and discuss their relation to computational complexity. The two measures reflect two different application scenarios: feeding one sentence at a time, and feeding as many words as possible into the GPU memory. The first scenario is designed to simulate, for example, instantaneous machine translation that translates text (or even speech) input from users. This is where current NAR models shine—we can make full use of parallelism across decoding positions in a GPU. For this reason, much prior work in NAR only measures speed using this metric (e.g., Gu et al., 2018; 2019b; Kasai et al., 2020; Li et al., 2020). The second scenario aims at a situation where we want to translate a large amount of text as quickly as possible. In this case, we see that AR models run faster than NAR models by a large margin. Computation at each time step is large enough to exploit parallelism in a GPU, which cancels out the benefit from parallel NAR decoding. Further, AR models can cache all previous hidden states (Ott et al., 2019) and compute each step in linear time complexity with respect to the sequence length. In contrast, NAR models necessitate a fresh run of quadratic self and cross attention in every decoding iteration.

Interestingly, using a deep encoder and a shallow decoder in NAR models fails to retain the original translation accuracy by using 6 layers each (§4.1). This suggests that departure from AR decoding necessitates more capacity in the decoder; the strategy is effective specifically for AR models. In particular, our analysis demonstrates that an NAR decoder requires more layers to learn target word ordering (§5). In summary, our contributions are the following:

- We challenge three conventional assumptions in NAR evaluation: suboptimal layer allocation, lack of distillation for AR baselines, and insufficiently general speed measures.
- We provide a complexity analysis and identify an optimal layer allocation strategy that leads to better speed-quality tradeoffs, namely a *deep-shallow* configuration.
- We perform extensive analyses and head-to-head comparisons of AR and strong NAR models on seven standard translation directions. We demonstrate that the accuracy gap between the two model families is much wider than previously thought and that NAR models are unable to capture target word order well without sufficiently deep decoders.

## 2 Reevaluating Non-Autoregressive Machine Translation

We critically examine in this section the evaluation practices and assumptions that are widely held in the non-autoregressive neural machine translation (NAR) literature (e.g., Gu et al., 2018; Ghazvininejad et al., 2019; Kasai et al., 2020). In particular, we focus on three aspects: speed measurement (§2.1), layer allocation (§2.2), and knowledge distillation (§2.3).

### 2.1 Speed Measures

One major benefit of NAR models over AR ones is their ability to generate text in parallel. Current research on measuring speed has focused solely on the setting of translating one sentence at a time where full parallelization is trivial with a single GPU. However, we argue that this speed measure is not realistic in some scenarios because the GPU memory is finite and the GPU unit in such a setting is underused. To address this issue, we use two translation speed metrics to measure inference speed:

- $S_1$ measures speed when translating one sentence at a time. This metric is used in standard practice and aligns with applications like instantaneous machine translation that translates text input from users immediately.
- $S_{max}$ measures speed when translating in mini-batches as large as the hardware allows. This corresponds to scenarios where one wants to translate a large amount of text given in advance. For instance, such large-batch machine translation is implemented in the Google cloud service.[1]

For all models, both metrics measure wall-clock time from when the weights are loaded until the last sentence is translated. We report speedups relative to an AR baseline with a 6-layer encoder and a 6-layer decoder following prior work (Gu et al., 2018; Li et al., 2020; Kasai et al., 2020).

---

[1] https://cloud.google.com/translate/docs/advanced/batch-translation.

## 2.2 LAYER ALLOCATION

Current evaluation practice in the NAR literature uses an equal number of layers for the encoder and decoder both in AR baselines and NAR models. However, previous studies in AR machine translation suggest that this allocation strategy leads to a suboptimal speed-quality tradeoff (Barone et al., 2017; Kim et al., 2019). These findings have several implications for evaluating NAR methods. We first discuss the strategy of *deep encoder, shallow decoder* (§2.2.1), and provide a theoretical analysis of the speed-quality tradeoff in the context of NAR evaluation (§2.2.2). Our analyses are verified empirically in the next section (§3).

### 2.2.1 DEEP ENCODER, SHALLOW DECODER

In line with prior work on deep encoders or shallow decoders (Barone et al., 2017; Wang et al., 2019a; Kim et al., 2019), we depart from the convention to allocate an equal number of layers on both sides and explore pairing a deep encoder with a shallow decoder for both AR and NAR methods. Here, we study the impact of such architectures and systematically compare AR and NAR methods.[2] As we will show in §3, an AR model with a *deep-shallow* configuration retains translation accuracy, but can substantially reduce decoding time. This is because at inference time, the encoder accounts for a smaller part of the overhead since its computation can be easily parallelized over source positions; on the other hand, the speedup gains from a lightweight decoder are substantial.

| | **By Layer** | | | **Full Model** | | |
|---|---|---|---|---|---|---|
| | Enc. | AR Dec. | NAR Dec. | AR $E$-$D$ | AR $E$-1 | NAR $E$-$D$ |
| **Total Operations** | $\mathcal{O}(N^2)$ | $\mathcal{O}(N^2)$ | $\mathcal{O}(TN^2)$ | $\mathcal{O}(EN^2 + DN^2)$ | $\mathcal{O}(EN^2 + 1 \cdot N^2)$ | $\mathcal{O}(EN^2 + DTN^2)$ |
| **Time Complex.** | $\mathcal{O}(N)$ | $\mathcal{O}(N^2)$ | $\mathcal{O}(TN)$ | $\mathcal{O}(EN + DN^2)$ | $\mathcal{O}(EN + N^2)$ | $\mathcal{O}(EN + DTN)$ |

Table 1: Analysis of transformers. Time complex. indicates time complexity when full parallelization is assumed. $N$: source/target length; $E$: encoder depth; $D$: decoder depth; $T$: # NAR iterations.

### 2.2.2 COMPLEXITY ANALYSIS

This section analyzes the complexities of transformer-based encoders, autoregressive and non-autoregressive decoders. We focus on two key properties: (1) the total amount of operations and (2) time complexity when full parallelization is assumed (Harris, 2007).

**Notation** For simplicity let us assume the source and target text have the same length $N$. $T$ is the number of iterations in an iterative NAR method (typically $T < N$). Let $E$ and $D$ denote the numbers of encoder and decoder layers.

Table 1 breaks down the comparison. AR and NAR models use the same encoder architecture. There are several interesting distinctions between AR and NAR decoders. First, although their total amounts of operations are both quadratic in sequence length, an NAR decoder with $T$ decoding iterations needs $T$ times more computation. Second, an AR decoder has time complexity quadratic in sequence length. This contrasts with the linear time complexity of an NAR decoder, which is the powerhouse of its $S_1$ speedup (§4.1). This is because the attention computation can be readily parallelized across target positions in NAR decoders.

By such comparisons we make the following key observations:

(a) For both AR and NAR models, the time complexity is dominated by decoders. When $T < N$, an NAR model has an advantage over its AR counterpart with the same layers.

(b) Innocuous as it may seem, the constant $T$ contributes major computational cost in terms of the total operations of NAR models. Empirically, $T$ needs to be at least 4 to perform competitively to AR models (Ghazvininejad et al., 2019; Kasai et al., 2020).

(a) suggests that one can significantly speed up $S_1$ decoding by using shallower decoders, while increasing the encoder depth only results in a mild slowdown. As we will show later in the experiments, AR decoders are much more robust to using fewer layers than NAR decoders. For example,

---

[2]Note that Kim et al. (2019) proposed other methods to optimize CPU decoding of AR models, but we do not apply them, to ensure fair comparisons between AR and NAR models.

AR $E$-1 can decode much faster than AR $E$-$D$ and comparably to NAR $E$-$D$, while retaining the accuracy of AR $E$-$D$. From (b), one may expect a different trend in $S_{max}$ from $S_1$: in large mini-batch decoding, an AR model can make use of the GPU's compute units, since now parallelization happens over the instances in a mini-batch. In other words, under the $S_{max}$ evaluation where the GPU is running close to its maximum flop/s, NAR can actually be slower since it needs more operations due to its iterative decoding. This is confirmed by our experiments (§4.1).

## 2.3 KNOWLEDGE DISTILLATION

Most NAR models rely on sequence-level knowledge distillation (Hinton et al., 2015; Kim & Rush, 2016) to achieve a reasonable speed-quality tradeoff, where NAR models are trained on output translations from a (larger) AR model. Nevertheless, standard practice in this area assumes that knowledge distillation is not required for the AR baseline. Here, we aim for fair evaluation by applying distillation to both model families; we depart from previous practice where NAR models trained *with* distillation are compared with AR models trained *without* (Ran et al., 2021; Sun et al., 2019; Shu et al., 2020; Zhou et al., 2020; Saharia et al., 2020) with a few exceptions (Ghazvininejad et al., 2019; Kasai et al., 2020). Our analysis (§5) demonstrates that AR models also benefit from knowledge distillation and that the accuracy gap between AR and NAR models is wider than previously established.

## 3 EXPERIMENTS

We compare NAR and AR models with different layer allocation strategies on standard machine translation datasets of varying languages and sizes. Our results show that deep-shallow AR models provide a better speed-quality tradeoff than NAR models.

## 3.1 BASELINES AND COMPARISON

Prior work has proposed various approaches to non-autoregressive machine translation (NAR). These methods must seek a balance between speed and quality: the more decoding parallelization is introduced into a model, the more the output quality deteriorates due to a conditional independence assumption. Some of the existing NAR models rescore the candidates with external autoregressive models (Sun et al., 2019; Li et al., 2020), or apply reordering modules (Ran et al., 2021). We mainly compare with two iterative NAR models (Ghazvininejad et al., 2019; Kasai et al., 2020) because of their strong performance *without* relying on any external system:

- **CMLM** (Ghazvininejad et al., 2019) predicts randomly masked target tokens given observed ones as well as the source. At inference time, it first predicts all target words non-autoregressively, and then iteratively masks and predicts the words that the model is least confident about. Following previous practice (Ghazvininejad et al., 2019; 2020b), we decode 5 candidate lengths in parallel (*length beam*) with $T = 4$ or $T = 10$ iterations.
- **DisCo** (Kasai et al., 2020) predicts every target token given an arbitrary subset of the rest of the target tokens. Following Kasai et al. (2020), we use their parallel easy-first inference, and set the maximum number of iterations to 10 and the length beam size to 5.

**Knowledge Distillation** We apply sequence-level knowledge distillation (Hinton et al., 2015; Kim & Rush, 2016) when training both NAR and AR models (§2.3). For the teacher models, we use left-to-right AR transformer models: transformer-large for EN-DE, EN-ZH, and EN-FR, and transformer-base for EN-RO (Ghazvininejad et al., 2019; Kasai et al., 2020).

## 3.2 EXPERIMENTAL SETUP

We experiment with 7 translation directions from four datasets of various training data sizes: WMT14 EN-DE (4.5M pairs, Bojar et al., 2014), WMT16 EN-RO (610K, Bojar et al., 2016), WMT17 EN-ZH (20M, Bojar et al., 2017), and WMT14 EN-FR (36M, EN→FR only). These datasets are all encoded into BPE subwords (Sennrich et al., 2016). We follow the preprocessing and data splits of previous work (EN-DE: Vaswani et al., 2017; EN-RO: Lee et al., 2018; EN-ZH: Hassan et al., 2018; Wu et al., 2019; EN-FR: Gehring et al., 2017). Following previous practice, we use SacreBLEU (Post, 2018) to evaluate EN→ZH performance, and BLEU (Papineni et al., 2002)

for others.[3] For all autoregressive models, we apply beam search with size 5 and length penalty 1.0. All models are implemented using `fairseq` (Ott et al., 2019). $S_1$ and $S_{max}$ wall-clock time speedups (§2) are evaluated on the same single Nvidia V100 GPU with 16GB memory. We apply half-precision training and inference (Micikevicius et al., 2018; Ott et al., 2019). It speeds up NAR models' $S_{max}$ by 30+%, but not $S_1$, in line with previous observations (Kim et al., 2019).

**Hyperparameters** We follow the hyperparameters of the base sized transformer (Vaswani et al., 2017): 8 attention heads, 512 model dimensions, and 2,048 hidden dimensions for both the encoder and decoder. For each model and dataset, the dropout rate is tuned from $[0.1, 0.2, 0.3]$ based on development BLEU performance. The EN→FR models are trained for 500K updates, while others for 300K (Kasai et al., 2020). Dev. BLEU is measured after each epoch, and we average the 5 best checkpoints to obtain the final model (Vaswani et al., 2017). See the appendix for further details.

## 4 RESULTS AND DISCUSSION

We provide in-depth results comparing performance and speedup across AR and NAR models.

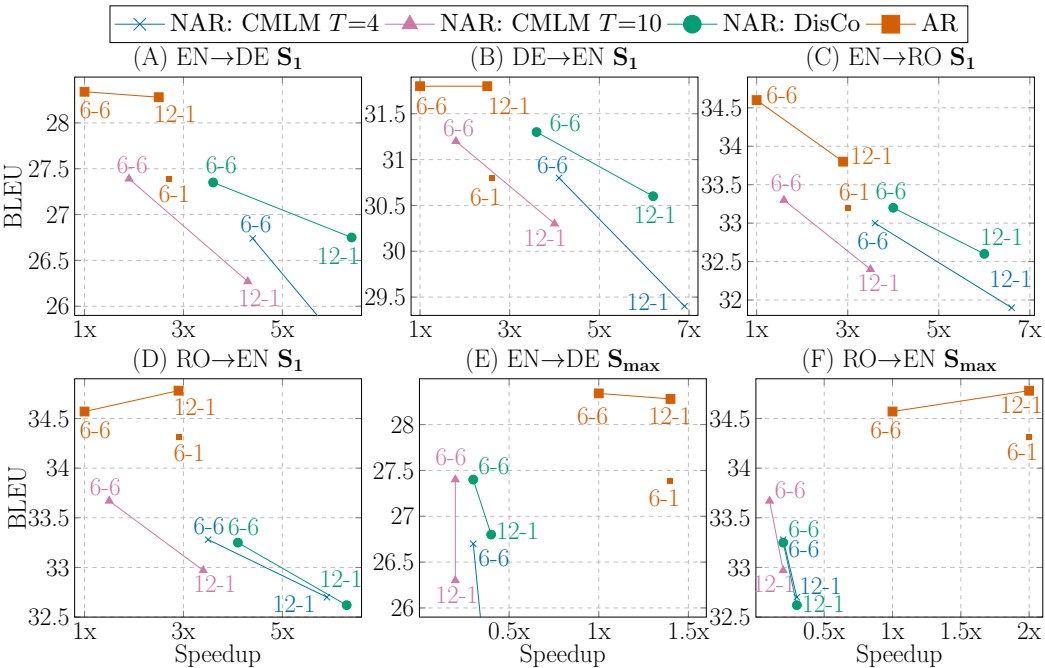

Figure 1: BLEU and speed comparisons with varying numbers of encoder and decoder layers on the test data. 12-1 denotes 12 encoder layers and 1 decoder layer. AR deep-shallow (12-1) finds a balanced middle ground in the tradeoff. Knowledge distillation is applied to all models (§3.1). See Table 5 in Appendix for more results.

### 4.1 DEEP ENCODER, SHALLOW DECODER

Fig. 1 shows translation speed-quality tradeoff curves of CMLM, DisCo, and AR models on WMT14 EN-DE and WMT16 EN-RO test data. For each model we plot the results of configurations with varying encoder and decoder depths. For brevity, we denote by $E\text{-}D$ a model with an $E$-layer encoder and a $D$-layer decoder. All speedups are measured relative to the AR 6-6 baseline (§2).

Firstly, under the 6-6 configuration, the AR model outperforms both CMLM and DisCo by a considerable margin in BLEU, but it achieves the slowest $S_1$ (see Fig. 1A–D). Using a single-layer decoder, AR 6-1 gains a substantial $S_1$ speedup (2.6x for EN→DE and 2.9x for RO→EN), but this comes at a cost of BLEU: 28.34 vs. 27.39 for EN→DE, and 34.57 vs. 34.31 for RO→EN. AR 12-1

---

[3]SacreBLEU hash: BLEU+case.mixed+lang.en-zh+numrefs.1+smooth.exp+test.wmt17+tok.zh+version.1.3.7.

lands on a balanced middle ground: it yields similar BLEU to AR 6-6, but its $S_1$ is more than 2.5 times faster. Notably, AR 12-1 achieves even faster $S_1$ than that of the CMLM 6-6 model with 10 iterations. In contrast, NAR 12-1 models generally suffer in BLEU compared to the 6-6 configuration; e.g., 26.75 (DisCo 12-1) vs. 27.35 (DisCo 6-6) in EN→DE.

Interestingly, all NAR models achieve slower $S_{max}$ than the AR 6-6 baseline (DisCo 6-6: 0.3x; CMLM 6-6 $T$=10: 0.1x in RO→EN). This is consistent with our complexity analysis in §2.2.2, where we found that with the same layer allocation, iterative NAR models need more total computation than the AR counterpart. AR 12-1 still gains a considerable speedup over AR 6-6 (2.0x in RO→EN). These results suggest that current NAR models have little advantage when translating a large amount of text given in advance, and one should clarify this distinction when discussing translation speed. See Table 5 in the appendix for full results from all four directions.

| Model | | | WMT17 EN→ZH | | | WMT17 ZH→EN | | | WMT14 EN→FR | | |
|---|---|---|---|---|---|---|---|---|---|---|---|
| | $T$ | $E$-$D$ | BLEU | $S_1$ | $S_{max}$ | BLEU | $S_1$ | $S_{max}$ | BLEU | $S_1$ | $S_{max}$ |
| CMLM | 4 | 6-6 | 33.58 | **3.5×** | 0.2× | 22.56 | **3.8×** | 0.2× | 40.21 | **3.8×** | 0.2× |
| CMLM | 10 | 6-6 | 34.24 | 1.5× | 0.1× | 23.76 | 1.7× | 0.1× | 40.55 | 1.7× | 0.1× |
| DisCo | | 6-6 | 34.63 | 2.5× | 0.2× | 23.83 | 2.6× | 0.2× | 40.60 | 3.6× | 0.2× |
| AR Deep-Shallow | 12-1 | | 34.71 | 2.7× | **1.7×** | **24.22** | 2.9× | **1.8×** | **42.04** | 2.8× | **1.9×** |
| AR | | 6-6 | **35.06** | 1.0× | 1.0× | 24.19 | 1.0× | 1.0× | 41.98 | 1.0× | 1.0× |
| Dist. Teacher | | 6-6 | 35.01 | – | – | 24.65 | – | – | 42.03 | – | – |

Table 2: Test BLEU and speed comparisons with varying numbers of encoder ($E$) and decoder ($D$) layers on large bitext. Best performance is bolded.

Table 2 presents results from large bitext experiments, EN↔ZH and EN→FR. We observe similar trends: AR deep-shallow achieves similar BLEU to AR 6-6 while boosting both $S_1$ and $S_{max}$ speed substantially. For EN↔ZH, AR deep-shallow has a more $S_1$ speedup than DisCo (2.7x vs. 2.5x in EN→ZH, 2.9 vs. 2.6 in ZH→EN). Particularly noteworthy is its performance in EN→FR: 42.04 BLEU, a 1.4 point improvement over the best NAR model. These results illustrate that the strategy of having a deep encoder and shallow decoder remains effective in large bitext settings, when the model has to learn potentially more complex distributions from more samples.

Lastly, Table 3 compares AR deep-shallow to recent iteration-based NAR results. All NAR models use the 6-6 configuration with the base size except that Imputer (Saharia et al., 2020) uses 12 self-attention layers over the concatenated source and target. Overall, our AR deep-shallow models outperform most NAR models, with the only exception being EN→RO where it underperforms Imputer by 0.6 BLEU points. However, each iteration takes strictly more time in the Imputer model than in CMLM or DisCo, since it requires a fresh run of 12-layer self attention over a concatenation of input and output sequences. As we saw in Fig. 1, AR deep-shallow yields comparable $S_1$ to CMLM 6-6 with 4 iterations, which would be about twice as fast as Imputer with 8 iterations.

## 4.2 CONSTRAINED VIEWS

In this section, we present two controlled experiments to compare NAR and AR models thoroughly.

**$S_1$ Speed Constraint** From §4.1 we see that compared to NAR models, AR deep-shallow yields a better translation speed-quality balance—despite being slightly slower in $S_1$ on some of the datasets, it achieves better BLEU across the board. To confirm this result, we further compare an AR deep-shallow model against two NAR models, controlling for $S_1$ speed. More specifically, we experiment with NAR models of varying encoder depths, and pair each with as many decoder layers as possible until it reaches AR 12-1's $S_1$ speed. Fig. 2 (left) shows the results. For CMLM $T$=4, CMLM $T$=10, and DisCo, the best configurations of 12-layer encoders were paired up with 12, 4, and 9 decoder layers, respectively. All NAR models improve performance as the encoder becomes deeper and surpass the scores of the 6-6 baselines (shown as squares along $x = 6$). Nonetheless, there is still a large BLEU gap from AR 12-1. This illustrates that the two NAR models are not able to match AR deep-shallow's accuracy under the same $S_1$ speed budget.

| Model | WMT14 EN−DE | | | | WMT16 EN−RO | | | | WMT17 EN−ZH | | | |
|---|---|---|---|---|---|---|---|---|---|---|---|---|
| | →DE | $T$ | →EN | $T$ | →RO | $T$ | →EN | $T$ | →ZH | $T$ | →EN | $T$ |
| CMLM | 25.9 | 4 | 29.9 | 4 | 32.5 | 4 | 33.2 | 4 | 32.6 | 4 | 21.9 | 4 |
| | 27.0 | 10 | 31.0 | 10 | 33.1 | 10 | 33.3 | 10 | 33.2 | 10 | 23.2 | 10 |
| LevT | 27.3 | >7 | – | – | – | – | 33.3 | >7 | – | – | – | – |
| DisCo | 27.3 | 4.8 | 31.3 | 4.2 | 33.2 | 3.3 | 33.2 | 3.1 | 34.6 | 5.4 | 23.8 | 5.9 |
| SMART | 27.0 | 4 | 30.9 | 4 | – | – | – | – | 33.4 | 4 | 22.6 | 4 |
| | 27.6 | 10 | 31.3 | 10 | – | – | – | – | 34.1 | 10 | 23.8 | 10 |
| Imputer | 28.0 | 4 | 31.0 | 4 | 34.3 | 4 | 34.1 | 4 | – | – | – | – |
| | 28.2 | 8 | 31.3 | 8 | 34.4 | 8 | 34.1 | 8 | – | – | – | – |
| AR 6-6 | **28.3** | $N$ | **31.8** | $N$ | **34.6** | $N$ | 34.6 | $N$ | **35.1** | $N$ | **24.2** | $N$ |
| AR 12-1 | **28.3** | $N$ | **31.8** | $N$ | 33.8 | $N$ | **34.8** | $N$ | 34.7 | $N$ | **24.2** | $N$ |
| Teacher | 28.6 | $N$ | 31.7 | $N$ | 34.6 | $N$ | 34.6 | $N$ | 35.0 | $N$ | 24.7 | $N$ |

Table 3: Test BLEU comparisons with iterative NAR methods. $T$ indicates the average # iterations. CMLM: Ghazvininejad et al. (2019); LevT: Gu et al. (2019b); DisCo: Kasai et al. (2020); SMART: Ghazvininejad et al. (2020b); Imputer: Saharia et al. (2020). Best performance is bolded.

**Layer Constraint** We can speed up autoregressive translation (AR) by developing a model with a deep encoder and a one-layer decoder. Here we thoroughly compare layer allocation strategies. Shown in Fig. 2 (middle) are results of NAR and AR methods under the constraint of 12 transformer layers in total. NAR models perform well when the decoder and encoder are balanced with slight tendency to deep encoders. On the other hand, the AR models perform consistently well with 4 or more encoder layers. This confirms that using deep encoders and shallow decoders is more effective in AR models than in NAR ones. Note that the number of parameters in each layer allocation differs since a decoder layer contains 30% more parameters than an encoder layer, due to cross attention.

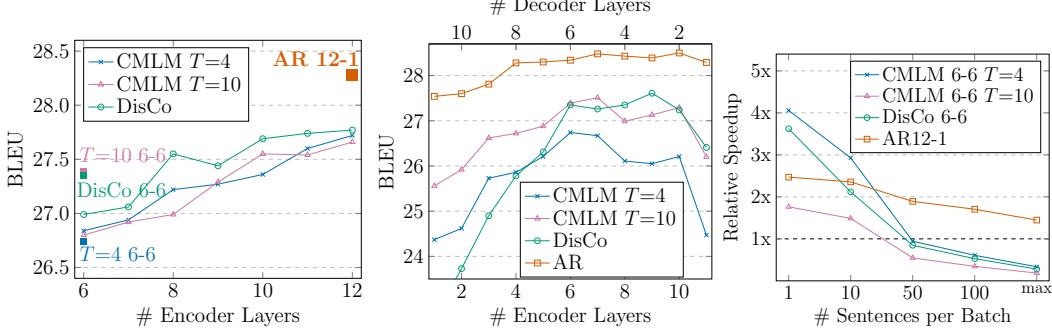

Figure 2: WMT14 EN→DE test results under various conditions. **Left**: varying depths of the encoder under the $S_1$ speed constraint of AR 12-1 ■. **Middle**: varying allocation of a total of 12 transformer layers over the encoder and decoder. **Right**: varying inference batch sizes.

## 5 FURTHER ANALYSIS

**Speedup and Batch Size** When decoding with large batches, NAR models can be slower than their AR counterpart (§4.1). Here we further study this effect. Fig. 2 (right) plots the relative speedups of different models' decoding with varying numbers of sentences per batch up to the hardware limit ("max," §2.1). The speedup by NAR models diminishes as the batch size grows: they have similar decoding speed to AR 6-6 with batch size 50, and become slower with larger batch sizes. In contrast, the speedup from AR deep-shallow decreases much more gradually.

**Decoder Depth and Reordering Words** From earlier results we see that NAR models need deeper decoders than AR models to perform well. We hypothesize that one reason is that NAR decoders need to learn to adjust to diverging word order between the source and the target: an AR decoder

| Model | E-D | Orig. | Reorder | Δ |
|-------|-----|-------|---------|---|
| CMLM, $T = 10$ | 6-6 | 27.4 | 31.7 | 4.3 |
| CMLM, $T = 10$ | 12-1 | 26.3 | 31.0 | 4.7 |
| DisCo | 6-6 | 27.4 | 31.0 | 3.6 |
| DisCo | 12-1 | 26.8 | 31.6 | **4.8** |
| AR | 6-6 | **28.3** | **32.6** | 4.3 |
| AR Deep-Shallow | 12-1 | **28.3** | **32.6** | 4.3 |

| Model | E-D | Raw | Dist. | Δ |
|-------|-----|-----|-------|---|
| CMLM, $T = 4$ | 6-6 | 22.3 | 25.9 | **3.6** |
| CMLM, $T = 10$ | 6-6 | 24.6 | 27.0 | 2.4 |
| Imputer, $T = 4$ | 12 | 24.7 | 27.9 | 3.2 |
| Imputer, $T = 8$ | 12 | 25.0 | 27.9 | 2.9 |
| DisCo | 6-6 | 24.8 | 27.4 | 2.6 |
| AR Deep-Shallow | 12-1 | 26.9 | **28.3** | 1.4 |
| AR | 6-6 | **27.4** | **28.3** | 0.9 |

Table 4: **Left**: WMT14 EN→DE test results in BLEU using reordered English input. **Right**: WMT14 EN→DE test results in BLEU that analyze the effects of distillation in fast translation methods. All distillation data are obtained from a transformer large. $E$: encoder depth; $D$: decoder depth; $T$: # iterations. Imputer (Saharia et al., 2020) uses 12 self-attention layers over the concatenated source and target, instead of the encoder-decoder architecture.

takes as input all preceding tokens and explicitly learns a conditional distribution, while an NAR decoder needs to learn target word ordering from scratch.

To test this hypothesis, we conduct the following controlled experiment in EN→DE translation. We choose German because of its divergence in word order from English. We first run the `fast_align` tool (Dyer et al., 2013)[4] on all bitext data (including the test set), and disable the NULL word feature to ensure that every English word is aligned to exactly one German word. We then shuffle the English words according to the order of their aligned German words. When multiple English words are aligned to the same German word, we keep the original English order. Finally, we apply the same BPE operations as the original data, and train and evaluate various models on the new reordered data. Table 4 (left) shows the results. AR gains the same improvement regardless of the layer configuration; in contrast, NAR 12-1 benefits more than NAR 6-6. This result supports our hypothesis that word reordering is one reason why NAR models need a deeper decoder.

**Effects of Distillation** We applied sequence-level knowledge distillation (Kim & Rush, 2016) to all models. Here we analyze its effects over the WMT14 EN→DE test data (Table 4 right). An AR transformer large model is used as the teacher model. All models benefit from distillation as indicated by positive Δ, including the AR models.[5] Many recent works only compare NAR models trained with distillation to AR models trained *without*. Our finding shows that that AR models with distillation can be an additional baseline for future NAR research. AR deep-shallow deteriorates much less on the raw data compared to the iterative NAR methods, suggesting that the strategy of speeding up AR models is better suited to modeling raw, complex data than the NAR methods.

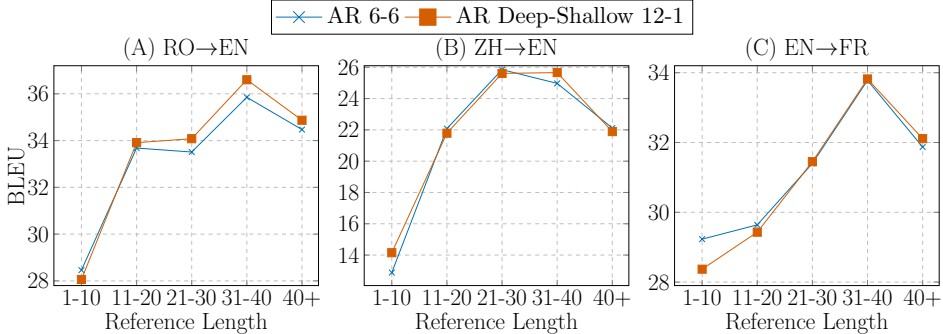

Figure 3: Test BLEU and target length comparisons for the AR 6-6 and deep-shallow 12-1 models.

---

[4] https://github.com/clab/fast_align.

[5] While the same distillation data are used in Ghazvininejad et al. (2019), they report a smaller improvement from distillation in BLEU. There are several potential reasons: we tuned the dropout rate for each model on the validation data and averaged the checkpoints that achieved the top 5 BLEU scores (§3.2). We note that our results are in line with the previous observations (Kim et al., 2019; Zhou et al., 2020; Kasai et al., 2020) where a similar improvement is gained by distilling an AR transformer large model to a base one.

**Breakdown by Sentence Length** Fig. 3 illustrates the relation between BLEU scores and reference translation lengths. We observe almost identical patterns between AR 6-6 and deep-shallow models, suggesting that they perform similarly regardless of the translation length.

**Can we reduce the decoder further?** We saw that an autoregressive model with a single-layer decoder and a sufficiently deep encoder can retain the accuracy of the baseline with 6 layers each. One may ask whether we can make the decoder even more compact. Our preliminary experiments showed that we can remove the feed-forward module from the decoder without hurting performance. This increases the $S_1$ speed by 10%. We leave further exploration to future work.

**Length Candidates and $S_{max}$ for NAR** Following the original works (Ghazvininejad et al., 2019; Kasai et al., 2020), we fixed the number of length candidates (i.e., the length beam size) to 5 for all NAR models, but a smaller beam size can speed up $S_{max}$ by allowing more sentences to be fed in a batch. Indeed, we found that NAR models can improve their $S_{max}$ by reducing the beam size at the expense of some accuracy drop. For example, we observed a loss of 0.5 BLEU points in EN→DE when decreasing the length beam size from 5 to 1. Nonetheless, NAR 6-6 models with beam size 1 still resulted in 0.6–0.9x $S_{max}$ compared to the AR 6-6 baseline.

## 6 FURTHER RELATED WORK

**Non-autoregressive Machine Translation** In addition to the work already discussed, several other works proposed to iteratively refine (or insert) output predictions (Mansimov et al., 2019; Stern et al., 2019; Gu et al., 2019a; Chan et al., 2019; 2020; Li et al., 2020; Guo et al., 2020). Other approaches include adding a light autoregressive module to parallel decoding (Kaiser et al., 2018; Sun et al., 2019; Ran et al., 2021), partially decoding autoregressively (Stern et al., 2018; 2019), rescoring output candidates autoregressively (e.g., Gu et al., 2018), mimicking hidden states of an autoregressive teacher (Li et al., 2019), training with different objectives than vanilla cross-entropy (Libovický & Helcl, 2018; Wang et al., 2019b; Shao et al., 2020; Tu et al., 2020; Saharia et al., 2020; Ghazvininejad et al., 2020a), reordering input sentences (Ran et al., 2021), training on additional data from an autoregressive model (Zhou & Keung, 2020), and modeling with latent variables (Ma et al., 2019; Shu et al., 2020). The approach of adding a light autoregressive module is closest to our method, but note that we pack all *non-autoregressive* computation into the encoder.

**Optimizing Autoregressive Transformer** Prior work has suggested various ways to optimize autoregressive transformers for fast inference. For example, Kim et al. (2019) considered shallow decoders and layer tying (Dabre & Fujita, 2019; Dehghani et al., 2019) on the transformer decoder and found that it sped up inference on CPUs, but not on a GPU, which was our focus. Kim et al. (2019) also explored concurrent streams where multiple batches are fed at once to make better use of a GPU. Shi & Knight (2017) proposed a vocabulary reduction method to speed up the last softmax computation. Senellart et al. (2018) also adopted vocabulary reduction and explored "fat decoder, thin encoder" on RNN-based models. Zhang et al. (2018) used dynamic programming in an average attention network to accelerate inference. Wu et al. (2019) developed a model with dynamic convolutions and compared its speed and accuracy with non-autoregressive models. Other works proposed methods to reduce attention computation in autoregressive transformers (Kitaev et al., 2020; Katharopoulos et al., 2020; Chelba et al., 2020; Peng et al., 2021). Some of these methods can be used orthogonally to further facilitate fast inference in a transformer, but our goal is to fairly reexamine the speed-quality tradeoff between autoregressive and non-autoregressive approaches under the same conditions.

## 7 CONCLUSION AND FUTURE WORK

We presented theoretical and empirical studies to demonstrate that autoregressive neural machine translation can be dramatically sped up by a simple layer allocation strategy: deep encoder, shallow decoder. Compared to strong non-autoregressive models, deep-shallow autoregressive models achieve substantial improvement in translation quality with comparable inference speed. Our results suggest that layer allocation, knowledge distillation, and speed measurement are important aspects that future work on non-autoregressive machine translation should take into consideration. More generally, a model with a deep encoder and a shallow decoder can be used for any sequence-to-sequence task, including large-scale pretraining (Lewis et al., 2020; Liu et al., 2020).

## ACKNOWLEDGMENTS

We thank Luke Zettlemoyer, Marcin Junczys-Dowmunt, Ofir Press, and Tim Dettmers as well as the anonymous reviewers for their helpful feedback and discussions on this work. This research was in part funded by the Funai Overseas Scholarship to Jungo Kasai. Nikolaos Pappas was supported by the Swiss National Science Foundation under the project UNISON, grant number P400P2_183911.

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

# A   APPENDIX

## A.1   RESULTS

| Model | $T$ | $E$-$D$ | →DE | $S_1$ | $S_{max}$ | →EN | $S_1$ | $S_{max}$ | →RO | $S_1$ | $S_{max}$ | →EN | $S_1$ | $S_{max}$ |
|---|---|---|---|---|---|---|---|---|---|---|---|---|---|---|
| | | | **WMT14 EN–DE** | | | | | | **WMT16 EN–RO** | | | | | |
| CMLM | 4 | 6-6 | 26.7 | 4.4× | 0.3× | 30.8 | 4.1× | 0.3× | 33.0 | 3.6× | 0.3× | 33.3 | 3.5× | 0.2× |
| | 10 | 6-6 | 27.4 | 1.9× | 0.2× | 31.2 | 1.8× | 0.2× | 33.3 | 1.6× | 0.1× | 33.7 | 1.5× | 0.1× |
| | 4 | 12-1 | 24.7 | **7.6×** | 0.4× | 29.4 | **6.9×** | 0.4× | 31.9 | **6.6×** | 0.3× | 32.7 | 5.9× | 0.3× |
| | 10 | 12-1 | 26.3 | 4.3× | 0.2× | 30.3 | 4.0× | 0.2× | 32.4 | 3.5× | 0.1× | 33.0 | 3.4× | 0.2× |
| DisCo | | 6-6 | 27.4 | 3.6× | 0.3× | 31.3 | 3.6× | 0.3× | 33.2 | 4.0× | 0.2× | 33.3 | 4.1× | 0.2× |
| | | 12-1 | 26.8 | 6.4× | 0.4× | 30.6 | 6.2× | 0.4× | 32.6 | 6.0× | 0.3× | 32.6 | **6.3×** | 0.3× |
| AR | | 6-6 | **28.3** | 1.0× | 1.0× | **31.8** | 1.0× | 1.0× | **34.6** | 1.0× | 1.0× | 34.6 | 1.0× | 1.0× |
| | | 6-1 | 27.4 | 2.7× | **1.4×** | 30.8 | 2.6× | **1.5×** | 33.2 | 3.0× | **2.0×** | 34.3 | 2.9× | **2.0×** |
| | | 12-1 | **28.3** | 2.5× | **1.4×** | **31.8** | 2.5× | 1.4× | 33.8 | 2.9× | **2.0×** | **34.8** | 2.9× | **2.0×** |

Table 5: Test BLEU and speed comparisons with varying numbers of encoder ($E$) and decoder ($D$) layers.

Table 5 provides comparisons of speed and quality in the WMT14 EN–DE and WMT16 EN–RO datasets.

## A.2   HYPERPARAMETERS AND SETTING

All of our models are implemented in `fairseq` (Ott et al., 2019) and trained with 16 Telsa V100 GPUs CUDA 10.1, and cuDNN 7.6.3. We used mixed precision and distributed training over 16 GPUs interconnected by Infiniband (Micikevicius et al., 2018; Ott et al., 2018). Apart from EN↔ZH where we used separate BPE operations, we tie all embeddings (Press & Wolf, 2017; Inan et al., 2017).

We generally follow the hyperparameters chosen in Vaswani et al. (2017); Ghazvininejad et al. (2019); Kasai et al. (2020) regardless of the numbers of encoding and decoding layers.[6] Specifically, we list the hyperparameters in Table 6 for easy replication. All other hyperparamter options are left as default values in `fairseq`.

---

[6]We use their code at https://github.com/facebookresearch/Mask-Predict and https://github.com/facebookresearch/DisCo.

| Hyperparameter | Value | Hyperparameter | Value |
|---|---|---|---|
| label smoothing | 0.1 | label smoothing | 0.1 |
| # max tokens | 4096 | # max tokens | 8192 |
| dropout rate | [0.1, 0.2, 0.3] | dropout rate | [0.1, 0.2, 0.3] |
| encoder embedding dim | 512 | encoder embedding dim | 512 |
| encoder ffn dim | 2048 | encoder ffn dim | 2048 |
| # encoder attn heads | 8 | # encoder attn heads | 8 |
| decoder embedding dim | 512 | decoder embedding dim | 512 |
| decoder ffn dim | 2048 | decoder ffn dim | 2048 |
| # decoder attn heads | 8 | # decoder attn heads | 8 |
| max source positions | 10000 | max source positions | 10000 |
| max target positions | 10000 | max target positions | 10000 |
| Adam lrate | $5 \times 10^{-4}$ | Adam lrate | $5 \times 10^{-4}$ |
| Adam $\beta_1$ | 0.9 | Adam $\beta_1$ | 0.9 |
| Adam $\beta_2$ | 0.98 | Adam $\beta_2$ | 0.999 |
| lr-scheduler | inverse square | lr-scheduler | inverse square |
| warm-up lr | $1 \times 10^{-7}$ | warm-up lr | $1 \times 10^{-7}$ |
| # warmup updates | 4000 | # warmup updates | 10000 |
| # max updates | 300K, 500K (EN→FR) | # max updates | 300K, 500K (EN→FR) |
| length penalty | 1.0 | | |

Table 6: Autoregressive (**left**) and non-autoregressive (**right**) `fairseq` hyperparameters and setting.

### A.3   SAMPLE OUTPUTS

Here we provide translation outputs randomly sampled from the validation data in ZH→EN. We do not find a qualitative difference between AR 6-6 and deep-shallow 12-1 models.

| Source | Reference | AR 6-6 | AR Deep-Shallow 12-1 |
|---|---|---|---|
| 上面所列出的当然不尽完整 | The previous list is not exhaustive, of course | The list above is certainly incomplete | The list above is certainly not complete |
| 一百万人，加拿大 总人口的十分之一，依靠政府的救济过活。 | One million people, a tenth of the entire Canadian population, were dependent on government relief. | One million people, one tenth of Canada's population, live on government aid. | One million people, one tenth of Canada's total population, depend on government aid for their survival. |
| 妇女企业家们还透过关于小企业管理的一系列培训课程得到援助,这种 培训课程最初于1994年1月在安曼新营开始。 | Women entrepreneurs were also assisted through a series of training courses in small-business management, first conducted in January 1994 at Amman New Camp. | Women entrepreneurs were also assisted through a series of training courses on small-scale enterprise management, which began in January 1994 at Amman New Camp. | Women entrepreneurs have also been assisted through a series of training courses on small enterprise management, which began at Amman New Camp in January 1994. |
| 事实上,必须在中期上保持对拯救生命和控制恐怖两方面所作的投入 ,以有效地扭转这一持续悲剧的势头。 | Indeed, the investment in both saving lives and reining in terror needs to be sustained over the medium-term in order to effectively turn the tide in this continuing tragedy. | In fact, the investment made in saving lives and controlling terror must be maintained in the medium term in order to effectively reverse the momentum of this continuing tragedy. | Indeed, investment in saving lives and controlling terror must be maintained in the medium term in order to effectively reverse the momentum of this continuing tragedy. |
| 这一复杂和动荡的世界恐怕是多元化、民主和自由的,在这世界上,美国正在力图剥夺我国作为一个主权国家 的合法位置,好象两国之间两百年的关系不算回事。 | In this complex and convulsed world that is supposedly pluralistic, free and democratic , the United States is trying to deny my country, Cuba, its rightful place as a sovereign nation. It is as if two centuries of relations between the two countries meant nothing. | I am afraid that this complex and volatile world is pluralistic, democratic and free, in which the United States is trying to deprive my country of its rightful place as a sovereign State, just as two hundred years of relations between the two countries are not worth it. | This complex and volatile world, which is feared, pluralistic, democratic and free, the United States is seeking to deprive our country of its rightful place as a sovereign State, as if two hundred years of relations between the two countries were not a matter. |

Table 7: Sample translation outputs from the ZH→EN validation data.