# OpenReview forum: "Deep Encoder, Shallow Decoder: Reevaluating Non-autoregressive Machine Translation"
_ICLR.cc/2021/Conference — ICLR 2021 Poster_

### Official Review · AnonReviewer3 · 2020-10-15
**Compelling argument that well-tuned AR methods are superior to current NAR models**

**Rating:** 7
**Confidence:** 3

**Review:**

Summary: Traditional NTM is done with a large encoder and large autoregressive (AR) decoder. Due to the sequential nature of the AR decoder, inference can be slow due to lack of parallelism (unless done at very large batch sizes). Non-Autoregressive (NAR) models have been proposed to alleviate this problem, but all NAR approaches trade off some translation quality for speed gains. In this paper, the authors claim that an alternative to NAR is to speed up standard AR decoding by reallocating network weights and layers to the (easily parallelizable) encoder, and making the decoder a single layer. They claim that this matches the speed of NAR models while keeping the performance of traditional AR models, making it a better choice in the design space than any NAR models. Comparisons are made to CMLM and DisCo NAR models to justify these claims with experimental evidence.

Pros:

1. The results are fairly compelling. A 12-layer encoder + 1-layer decoder model matches the 6/6 model in translation quality, and, as promised, speedups are comparable to the speedups obtained from NAR methods.

2. The point made about knowledge distillation is also valuable. Knowledge distillation is valuable tool for speeding up models, and it is just as applicable to AR speedups as NAR speedups.

Cons:

1. In my view, the comparison of S_{max} is misleading. According to the authors, it "measures speed when translating in mini-batches as large as the hardware allows. This is closer to practical scenarios where one wants to translate a large amount of text". However, this is emphatically *not* the purpose of NAR models, which are *specifically* for the low-batch-size case. In addition, I disagree that this "is closer to practical scenarios"; many user queries are in fact short sentences or phrases, rather than many paragraph articles. Both use cases exist, and comparing NAR vs AR models on S_{max} is comparing a slightly wrong metric.

This could be improved by instead plotting speed as a function of batch size, rather than choosing arbitrarily the two extremes. A curve of batch size vs speedup is valuable, as done in Fig 2.

2. I found that Section 2.2.1 and 2.2.2, formal derivations of runtime and number of operations, were unnecessary. To me, these formalisms made the paper harder to understand and parse, rather than easier.

This could be improved by reducing these to key points: (a) encoders are parallelizable (b) T is a significant factor in NAR models, and is around 4-10 for good performance (c) AR decoders are more robust to using fewer layers than NAR decoders, etc.

3. Figure 1 is confusing.

I think this could be improved by converting it into a table with a "NAR" section and an "AR" section, and each row being a model.

Recommendation: Accept. Although I had some criticisms of the paper, they pertained mostly to clarity, style, and precise definition of chosen metrics; the key results of the paper are solid.

---

> ### Author Response · Authors · 2020-11-18
> **Response to Reviewer 3**
>
> Thank you for your review and suggestions to improve the clarity and presentation of the paper.
>
> Reviewer 3 raises a concern about our S_max speed measure. We updated our description: S_max corresponds to scenarios where a large amount of text is given in advance to translate (see also our response to Reviewer 1). As the reviewer points out, S_1 and S_max correspond to different use cases. We also share the intuition that most NAR models are specifically suitable for small batch translations at present. However, we suspect that large batch translation is equally critical in applications. For example, [Google](https://cloud.google.com/translate/docs/advanced/batch-translation) and [Amazon](https://aws.amazon.com/about-aws/whats-new/2020/01/amazon-translate-introduces-batch-translation/) provide large-batch neural machine translation services that are used when translating large collections of text or HTML documents. It will be impactful if future NAR systems can achieve significant speedups in S_max. We added this discussion to the paper.
>
> Reviewer 3 suggests a way to improve the clarity of Fig. 1. We chose the visual illustration (BLEU vs. speedup) to show how speed-quality tradeoffs differ from model to model and where each configuration sits in the spectrum. Following Reviewer 1’s suggestion, we also added results from more translation directions, which might facilitate clarity.
>
> The reviewer raises a concern about the clarity of our complexity analysis in Sec. 2.2. We agree that the core points that we learn from the theoretical analysis are: (a) encoders are parallelizable over source positions and the time complexity is dominated by decoders, and (b) T contributes major computational cost in terms of the total operations of NAR models. These observations provide a basis for our empirical results from a theoretical perspective. We modified the section in the updated version and will consider making it more compact for better readability.

---

### Official Review · AnonReviewer2 · 2020-10-27
**Solid experiments but more insight is required.**

**Rating:** 5
**Confidence:** 4

**Review:**

** Summary **

In this paper, the authors present a throughout study on the deep-shallow architecture for autoregresstive machine translation model (AR) and non-autoregressive model (NAR). Authors compare the two types of models from speed measures, layer allocation and the usage of knowledge distillation.
Many results in this paper are interesting and meaningful:
(1)	Deep-shallow architecture (in this paper, 12-1) can achieve good BLEU performances across different tasks and speed up inference.
(2)	NAR is not as fast as AR when using S_{max} evaluation (i.e., the inference in batch model, where all GPU memory should be utilized).
(3)	Knowledge distillation is helpful for both AR and NAR.

The experiment results are solid and reproducible.
The paper is well written.

** Significance **
Although this is a well-written paper with solid results, I still have the following concerns:
1.	To speed up inference, one can easy get to use a shallow decoder to reduce inference time, since the decoding process in AT takes most of the time. The results do not give me so much surprise.
2.	Why the 12-1 architecture can achieve better results than 6-6 architecture on AR, but worse results on NAR? (see Figure 1 (B)(C)) What roles do encoder and decoder play in different modes (i.e., AR and NAR?) The following terms might be helpful to you:
a.	The comparison of their training/validation curves to verify whether the improvement is due to better training/generalization.
b.	In Figure 1, the results are mainly obtained on 6-1, 12-1, 6-6. What if we try more combinations?
i.	Fix the encoder layer as 6/12 and vary the decoder layers {1,2,4};
ii.	Fix the decoder layer as {1,2,4} and try more the encoder layers?
In this way, we can see how encoder/decoder layer effect the results (speed and quality).
3.	What are the differences between the output of 6-6 and 12-1? Any human analysis? I think you can briefly summarize the discovery in the paper with examples attached in the appendix. BLEU is important but you should evaluate your model from more perspectives.

---

> ### Author Response · Authors · 2020-11-18
> **Response to Reviewer 2**
>
> Thank you for your review and interesting suggestions for further analysis of our results.
>
> The authors share the reviewer’s intuition that a shallow decoder should reduce inference time when translating one sentence at a time (S_1). This is also consistent with our complexity analysis in Sec. 2.2.2. However, it was surprising to us that increasing encoder layers can cancel out the expected accuracy drop from a one-layer decoder. Encoders and decoders seem inherently different as decoders are (conditional) language models for the target language, but our results suggest that some encoder and decoder layers are interchangeable.
>
> Reviewer 2 asks about why the deep-shallow configuration harms the accuracy of NAR models unlike AR ones. We also find this difference between the two families interesting and aimed to answer this question in our analysis (Sec. 5). We hypothesized and confirmed that a one-layer decoder in NAR models fails to capture target word ordering with the lack of sequential dependencies (see Table 4 left for the results from our controlled experiment). The reviewer suggested two additional angles for more analysis: a) comparing training and validation performance and b) varying the decoder depth while fixing the encoder depth and vice versa.
>
> - a) In our original experiments we saw AR 12-1 models achieved better training and validation losses than AR 6-1 models, suggesting that the improvement from a deep encoder comes from better training, not just better generalization.
> - b) While we designed our experiments under our computational budget, Fig. 2 middle is related to the suggested experiments. We fixed the total of 12 layers and varied the allocation between the encoder and the decoder. The figure shows that NAR models perform best when the decoder and encoder depths are balanced.  On the other hand, the AR models perform consistently well with 4 or more encoder layers. This again confirms our hypothesis that NAR models need a deeper decoder for word reordering than AR models.
>
> We think that why a deep-shallow configuration works in autoregressive machine translation is an interesting and important question that can have broad impacts on sequence generation tasks beyond machine translation. Our analysis shows that word reordering is one of the key factors that differentiate NAR and AR models, but more extensive analysis is left for future work.
>
> We also agree that more analysis or evaluation of translation outputs will be useful. We added to Sec. 5 a breakdown of performance based on the sentence length and confirmed that a shallow decoder does not particularly suffer in long sentence translation. We also randomly sampled translations on the validation data where AR 12-1 and 6-6 models differ and reported them in the appendix of the revision.  From the limited samples, we did not find qualitative differences between AR 12-1 and 6-6 outputs. We also think that large-scale human evaluation adds valuable assessment to automatic metrics such as BLEU. This paper focuses on revisiting the current speed-quality evaluation practice for NAR machine translation, but we hope that future work will explore deep-shallow configurations on more machine translation benchmarks, such as WMT and WAT competitions, where human evaluation is extensively performed.

---

> > ### Comment · AnonReviewer2 · 2020-11-22
> > **Further discussion**
> >
> > Thanks the authors for the reply. The results are indeed interesting, where the 12-1 architecture can achieve almost the same results as 6-6, with much faster speed. And the 12-1 architecture achieves faster speed than NAR.
> > Finally, I have the following questions to discuss with the authors:
> >
> > 1.	In Section 3.1, last paragraph “knowledge distillation”, you mentioned that you used distillation to all models.
> > (1-A) That is, in all your experiments, results in Figure 1,  Table 2, Table 3, Table 4, Figure 3, knowledge distillation is leveraged, am I right? Perhaps I miss something, but will a deep-shallow encoder (specifically, 12-1) work when not applying distillation ?
> > (1-B)  In section 3.1, you pointed that “For the teacher models, we use left-to-right AR transformer models: transformer-large” Please report the performances of all teachers models.
> >
> > 2.	Currently, I do not find the curves for “ ... In our original experiments we saw AR 12-1 models achieved better training and validation losses than AR 6-1 models ...” Could you please point out where are they? And if they are not attached, could you please attach them in the appendix? For example, I want to see the training/validation curves of the models in Figure 1(A), which could be easily extracted from the training log. Also, please kindly point out whether distillation is used for each model.
> >
> > 3.	Currently, you only work on transformer_base setting, where the embedding dimension and hidden dimension of FFN layers are 512 and 2048. What if we work on the transformer big setting? I know the question is a little late but you can provide some preliminary results if any.
> >
> > ** One more suggestion:
> > In the author response, the authors claim that “...... but our results suggest that some encoder and decoder layers are interchangeable”, and I notice the Figure 2 (middle).  I think in the future, the authors need to find more evidence to support this claim but currently, this statement is not quite correct, since when the architectures range from 1-12 to 6-6, the results vary a lot.  A possible experiments is that: fix the number of encoder (decoder) layers and gradually increase decoder (encoder) layers. In this way, we can see how much additional improvement can be obtained by introducing one more layer, and find the relation/differences between encoder and decoder layers.

---

> > > ### Author Response · Authors · 2020-11-24
> > > **Response to Reviewer 2 for Further Discussion**
> > >
> > > Thank you for your follow-up comments and questions on our response.
> > >
> > > 1. The reviewer’s understanding is right; we applied knowledge distillation to all models presented in Figs. 1-3, Tables 1-3, and Table 4 (left). Knowledge distillation is necessary in almost all NAR models in the literature to achieve competitive accuracy to autoregressive baselines, and we aimed to compare AR and NAR models fairly in this work. Table 4 (right) compares the difference between the performance *with* and *without* knowledge distillation for Imputer, CMLM, DisCo, AR Deep-Shallow 12-1, and AR 6-6. All models benefit from distillation, but as the reviewer points out, AR 6-6 and AR 12-1 are much less affected than the others. Following the reviewer’s suggestion, we added the performances of all teacher models in Tables 2 and 3.
> > >
> > > 2. We do not have access to the detailed training logs from our original experiments at this point, and we want to rerun training to plot training and validation loss curves. We will rerun those experiments presented in Fig. 1 and add plots and a discussion to the appendix of the final version. We clarified that knowledge distillation is applied to all models in the caption of Fig. 1 in the latest revision, following the reviewer’s suggestion.
> > >
> > > 3. We have not experimented with the transformer large setting. Most of the prior works in NAR machine translation use base-sized transformers, and we wanted to extensively compare AR and NAR models under our computational budget over varying configurations. Nonetheless, we agree that results from scaling up the model size would be useful especially for applications. If time permits, we will run experiments with the large setting and report findings in the final version.
> > >
> > > 4. Regarding Fig. 2 (middle), we can observe that the BLEU score of the AR system changes relatively little compared to the NAR systems (especially with 4 encoder layers or more). This finding supported our intuition that **some** (certainly not all of them because AR 1-11 underperforms AR 6-6 by about 1 BLEU point as the reviewer mentions) and decoder layers in AR are interchangeable. This is not the case with the NAR systems. We agree with the reviewer that fixing the encoder (decoder) depth while varying the decoder (encoder) depth will be interesting additional experiments to further analyze the impacts of encoder and decoder layers. We will conduct these experiments and add results to the final version.

---

### Official Review · AnonReviewer4 · 2020-10-28
**detailed analysis of current shortcomings in non-autoregressive MT research**

**Rating:** 7
**Confidence:** 3

**Review:**

The authors advocate for fair comparison between autoregressive (AR) and non-autoregressive models (NAR) in non-autoregressive machine translation (NAT) research. They highlight three main aspects where the comparison has not been fair so far in the literature - suboptimal layer allocation, insufficient speed measurement, and lack of knowledge distillation. They perform extensive comparisons between AR and NAR models in these 3 aspects and report interesting results.

Pros:
- The paper is very well written and easy to follow. Limitations of previous works have been well explained, and the motivation behind their work is clearly evident. The experiments are rigorous.
- The paper shows that autoregressive models with deep encoders and shallow decoders are comparable and sometimes outperform current state of the art iterative semi-autoregressive models in terms of quality-latency tradeoff. I believe these findings can be valuable to the non-autoregressive MT and machine translation community in general.

Cons / Suggestions:

- When doing the S_{max} speed comparisons, I am assuming that the maximum amount of tokens fitting in the GPU memory also depends on the length beam. I will suggest the authors to clarify this, and possibly include an experiment showing what speed-quality tradeoff iterative models achieve with different values of length beam.
- In Section 2.3, the authors should point out that some previous works like Mask-Predict (Ghazvininejad et al., 2019) do report AR baselines with knowledge distillation.
- Unless I inferred the results incorrectly, Mask-Predict finds almost no improvement in AR baselines when using distilled data, while in this paper, the authors find an improvement of around 1 BLEU point. Can the authors comment on why this could be the case? Can this  be because of variations in the distilled data used, or some difference in model training. I believe that 1 BLEU point is too big a difference to be accounted for as a noise signal.

Overall, this paper proposes a different direction for research in fast and accurate machine translation, and points out the current shortcomings of research in non-autoregressive machine translation in details. I find findings by this paper very interesting and potentially valuable to the concerned research community, hence I recommend its acceptance.

---

> ### Author Response · Authors · 2020-11-18
> **Response to Reviewer 4**
>
> Thank you for your review and interesting questions about our experiments.
>
> Reviewer 4 asks about length beam size (# length candidates) and its implication on S_max for the iterative NAR models. We used a fixed length beam size of 5 for both the CMLM and DisCo models following [Ghazvininejad et al., (2019)](https://arxiv.org/abs/1904.09324) and [Kasai et al., (2020)](https://arxiv.org/abs/2001.05136) as noted in Sec. 3.1.  We agree with the reviewer that a small length beam can speed up NAR’s S_max by allowing more sentences to be fed in a batch. We subsequently ran experiments and confirmed that NAR can improve its S_max at the expense of some accuracy drop (e.g., a loss of 0.5 BLEU points in EN->DE when reducing the length beam size from 5 to 1). Reducing the length beam size is an effective strategy to improve NAR’s S_max, but it still underperforms the AR baseline both in S_max speed and accuracy. NAR 6-6 models with beam size 1 resulted in 0.6x-0.9x S_max compared to the AR 6-6 baseline. This is consistent with our results that NAR models with beam size 5 yield about 0.1-0.2x S_max of AR 6-6 (Fig. 1 and Table 2) because a model with length beam size 1 can translate about 5 times more sentences in each batch than length beam size 5. We added this finding to the new version.
>
> Reviewer 4 points out that previous works like Mask-Predict ([Ghazvininejad et al., 2019](https://arxiv.org/abs/1904.09324)) report AR baselines with knowledge distillation. While our paper identifies problems with dominant evaluation practice in NAR machine translation, we absolutely need to acknowledge this point. We updated the paper accordingly.
>
> The reviewer asks about the discrepancy between our AR distillation results and those reported in [Ghazvininejad et al., (2019)](https://arxiv.org/abs/1904.09324). We used the same distillation data for the overlapping language pairs. We suspect that there are several possible reasons for this difference. Firstly, we tuned our dropout rate from [0.1, 0.2, 0.3] for each dataset and each model on the validation data, but [Ghazvininejad et al., (2019)](https://arxiv.org/abs/1904.09324) fixed the rate to 0.3 for all translation directions. Secondly, we obtained the final model by averaging the checkpoints that achieved the top 5 BLEU scores for every model. More details of our training setups are described in Sec. 3.2 and Appendix A.2. We note, however, that our results are consistent with [Zhou et al., (2020)](https://arxiv.org/abs/1911.02727) (Fig. 4), [Kasai et al., 2020](https://arxiv.org/abs/2001.05136) (Tables 1-3), and [Kim et al., (2020)](https://www.aclweb.org/anthology/D19-5632) (Table 1) where they saw a similar improvement in BLEU by distilling an AR transformer large model to an AR transformer base model. We added this discussion in the updated version.

---

> > ### Comment · AnonReviewer4 · 2020-11-21
> > **Response to Authors**
> >
> > I thank the authors for responding to my concerns and questions. I am satisfied with their response, and keep my score unchanged.

---

### Official Review · AnonReviewer1 · 2020-11-03
**Deep encoders and shallow decoders for NMT**

**Rating:** 9
**Confidence:** 5

**Review:**

Summary:
The paper proposes deep encoder and shallow decoder models for auto-regressive NMT. They compare rigorously to NAR models. They also study three factors: layer allocation, speed measurement and knowledge distillation. They include that with a 12E-D1 model they obtain significant speed-up and can outperform the standard 6-6 AR model and almost always beat the NAR model in terms of quality. They also show that NAR models need deep decoders because they need to handle reordering.

Reasons for score:
I scored this paper a 9. I think this is an important paper which establishes very strong AR baselines for future NAR work in the field. They correctly point out the three issues with the comparisons that many NAR papers make. They conduct various meaningful ablation studies and validate their various hypothesis properly. They also show that certain factors like knowledge distillation need to be applied to both AR and NAR systems. Finally, they advocate for reporting both S_1 and S_max when comparing speed gains.

Cons:
- One issue I had with the presentation of the results was the selection of different formats and language pairs for different experiments. For example, table 2, 3 and 4 report on different subsets of language-pairs. Same with the figures. This might raise questions of whether the authors are randomly subselecting or selecting favorable subsets. I would have liked to see all experiments done on all LPs.


Minor comments:
- Section 2.1: S_max - "This is closer to practical scenarios where one wants to translate a large amount of text." - this is a very subjective statement and I would tone this down.
- Section 2.2.2: "Denote respectively by E and D the numbers of encoder and decoder layers." -- please fix grammar

Missing citations:
- Section 1: Along with Sutskever, Bahdanau and Vaswani. please also cite https://www.aclweb.org/anthology/D13-1176.pdf and Wu et al. 2016 (https://arxiv.org/abs/1609.08144) when you mention state-of-the-art NMT.

---

> ### Author Response · Authors · 2020-11-18
> **Response to Reviewer 1**
>
> Thank you for your review and suggestions to improve our paper.
>
> Reviewer 1 raises a concern regarding the selections of language pairs and translation directions in our experimental results. Since we found similar patterns in the other translation directions, we reported them in the appendix for space in the initial submission. For the updated version, we added three more plots to Fig. 1; we now cover all translation directions with Fig. 1 and Table 2. For the word reordering experiment to analyze the different effects of a shallow decoder in AR and NAR models (Table 4 left), we chose German because of its diverging word order from English. We clarified this point in the updated version.
>
> Following the reviewer’s suggestion, we softened our comment on S_max: S_max corresponds to scenarios where a large amount of text is given in advance to translate.
>
> Thanks for pointing to these related works. We have added them in the revision.

---

> ### Comment · AnonReviewer1 · 2020-11-23
> **Response to authors**
>
> Thank you for addressing my feedback. The updated paper has definitely improved. I leave my score unchanged.

---

### Decision · Program_Chairs · 2021-01-07
**Final Decision**

**Decision:**

Accept (Poster)

**Comment:**

This work demonstrates that autoregressive (AR) models for machine translation can can be competitive with their non-autoregressive (NAR) counterparts in terms of practicality. This is a timely observation, given the flurry of recent work on NAR models, whose primary benefit is often cited to be fast inference.

It was argued that the results are not surprising -- if this is the case, I still think this work merits acceptance because its thesis runs counter to the direction the field as a whole seems to be moving in, and the results are convincing. That said, I agree with the authors that the observation that some encoder and decoder layers are interchangeable, is not self-evident (i.e. it _is_ surprising). This is of course subjective to some degree, so I am making a judgement call here. The work also has value in that it draws attention to some practices regarding evaluation in NAR machine translation literature that could be improved and made more fair (specifically regarding comparison with AR models).

There were some concerns about whether these models should be evaluated in the small-batch or large-batch setting. The authors have updated their manuscript in response, and it now explicitly discusses both settings. The authors have also run more experiments and added several additional results requested by reviewers to the manuscript.

All things considered, I am inclined to follow the majority and recommend acceptance.